

# Moulting behaviour in the trilobite *Oryctocephalus indicus* (Reed, 1910) from the Cambrian Miaolingian Series (Wuliuan Stage) of Jianhe, South China

Shengguang Chen[1,2,3], Xinglian Yang[1,2], Xiong Liu[1], Zhengpeng Chen[1], Zhixin Sun[3,4] and Fangchen Zhao[3,4]

[1] College of Resource and Environmental Engineering, Guizhou University, Guiyang, China
[2] Key Laboratory of Karst Georesources and Environment (Guizhou University), Ministry of Education, Guiyang, China
[3] State Key Laboratory of Palaeobiology and Stratigraphy, Nanjing Institute of Geology and Palaeontology of the Chinese Academy of Sciences, Nanjing, China
[4] College of Earth and Planetary Sciences, University of Chinese Academy of Sciences, Beijing, China

Corresponding authors
Xinglian Yang, yangxinglian2002@163.com
Fangchen Zhao, fczhao@nigpas.ac.cn

## ABSTRACT

The accurate interpretation of trilobite moulting behaviour relies on a comprehensive understanding of their moult configurations, yet the focus has commonly been limited to a brief description of the exuviae, and how differences in moulting behaviour further impact the preservation of exuviae is often ignored. This study investigates the configuration, style, and process of moulting in *Oryctocephalus indicus* through analysis of 88 exuviae collected from the Kaili Formation (Cambrian, Wuliuan) in Guizhou Province, South China. The moult configurations of *O. indicus* are typically characterised by the lower cephalic unit (LCU), which comprises the librigenae and rostral-hypostomal plate connected as a whole, detached from the cephalon and positioned anterior to the thoracopygon, while the cranidium is mostly absent. From detailed observation and description of the available material, we believe that *O. indicus* completes its moult through an exuvial gape formed by disarticulation of the facial sutures, rostral sutures and/or sutures of the cephalothoracic joints. Although many exuviae exhibited an opening at the cephalothoracic joint—disjunction of which is usually accompanied by disarticulation of both the facial and rostral sutures—the Salter's configuration produced by the 'Salterian' mode of moulting was not observed. Additionally, the structural characteristics of Henningsmoen's configuration, Harrington's configuration, and Somersault's configuration are discussed based on the exuviae of *O. indicus*, and Henningsmoen's configuration has been categorised into three types according to the different states of fossil preservation. In this article, apart from promoting further research on moulting behaviour in *O. indicus*, we also provide a supplement for moult configuration based on the exuviae, which offers new materials for studying moulting behaviour in oryctocephalid trilobites.

**How to cite this article** Chen S, Yang X, Liu X, Chen Z, Sun Z, Zhao F. 2023. Moulting behaviour in the trilobite *Oryctocephalus indicus* (Reed, 1910) from the Cambrian Miaolingian Series (Wuliuan Stage) of Jianhe, South China. *PeerJ* 11:e16440
http://doi.org/10.7717/peerj.16440
## INTRODUCTION

In a similar way to modern arthropods, trilobites shed their exoskeletons periodically during their ontogenetic process (*Henningsmoen, 1975*; *Miller & Clarkson, 1980*; *McNamara & Rudkin, 1984*; *Whittington, 1990*; *Drage & Daley, 2016*; *Drage, 2019*). Comprehending the moulting behaviour and processes of trilobites is imperative, as they played a pivotal role in ancient marine ecosystems with intricate evolutionary histories shaped by their ecdysial capability (*Daley & Drage, 2016*). Moulting is thought to be an opportunistic behaviour for trilobites, which employed various strategies for moulting (*Miller & Clarkson, 1980*; *Brandt, 2002*). The diversity and predominance of moult configurations are complex, although previous studies have shown that this variability is not well related to the actual phylogenetic relationships of the species (*Budil & Bruthansová, 2005*), this variability does not seem to be fully summarised in terms of overall differences in body morphology alone. Moulting behaviours, moulting movements during exuviation, and moulting configurations show significant variability among trilobites (*Drage, 2019*); each study on trilobite moulting replenishes a larger aggregation of occurrences and behavioural descriptions that can be analysed to interpret evolutionary trends for the group (*Drage & Daley, 2016*).

*Oryctocephalus indicus Reed (1910)*, a small, cosmopolitan oryctocephalid trilobite with a wide distribution range across South China, India, western North America, and Siberia (*Zhao et al., 2019*). The first appearance of this species defines a conterminous base of the global Cambrian Miaolingian Series and Wuliuan Stage (*Zhao et al., 2019*); thus, it is necessary to deepen our understanding of the Cambrian trilobite taxa through a detailed study on their moulting behaviour. As one of the exceptionally preserved Konservat Lagerstätte worldwide, the Kaili Biota (Wuliuan Stage) in Guizhou Province has yielded numerous trilobite fossils and exuviae, providing excellent materials for studying the moulting behaviour of trilobites (*Chen et al., 2021*; *Chen et al., 2022*). Previous research on trilobite moulting mainly focused on specimen descriptions, only a few publications discussing the moulting process (*e.g.*, *McNamara & Rudkin, 1984*; *Speyer, 1985*; *McNamara, 1986*; *Whittington, 1990*; *Rustán et al., 2011*; *Drage et al., 2018*). *Chen, Han & Zhao (2008)* provided some explanations for the moulting behaviour of *O. indicus*, confined to a description of moult configurations, thus more materials are needed to further explore the moulting style and process. This article provides a comprehensive description on the moulting configuration and style of 88 newly collected exuviae specimens from *O. indicus*, which elucidate the moulting process and complement the investigation of moulting behaviour in *O. indicus*.

## GEOLOGICAL SETTING

The Kaili Formation in the study area is primarily composed of silty mudstone, calcareous mudstone, and a set of terrigenous clastic rocks dominated by shale, with developed horizontal bedding and few ripple marks and bioturbated structures (*Zhou et al., 1979*; *Yin, 1987*). Previous research has indicated that the Kaili Formation in Jianhe, Guizhou Province, was deposited within a tropical slope transition sedimentary environment between the Yangtze platform (carbonate deposition) and Jiangnan Basin (fine-grained deposition)

(*Peng & Babcock, 2001*). The Kaili Formation is deposited under low-energy conditions below the storm wave base, where gravity driven suspensions of fine-grained sediment were set up by storm wave disturbance of substrates lying in shallower waters upslope (*Gaines et al., 2011*). In addition, the study of buried facies and biological preservation characteristics indicated that the Kaili Formation had weak hydrodynamic force during deposition, and the redox interface was close to the seafloor, where bioturbation and scavenging behaviour rarely occur due to the absent or poorly developed infaunal activity (*Zhang et al., 1996*). At the same time, the fine clay deposits protected the intact preservation of organisms enclosed in the mud, and the early diagenetic mineral (pyrite) plays a crucial role in preventing the soft body from rotting and replicating the soft tissue (*Zhu, Erdtmann & Zhao, 1999*). In summary, the Kaili Formation was deposited in a low-energy environment within the slope transition zone. In arthropods, exoskeletons can become disarticulated due to decay or physical processes, and the intra-specific variability of moult configurations can only be preserved under exceptional in-situ preservation conditions (*Drage et al., 2018*). The weak biological disturbances and favourable diagenetic conditions in the Kaili Formation provided an excellent setting for preserving trilobite exuviae. The *Oryctocephalus indicus* specimens used in this study were obtained from the Kaili Formation of the Miaobanpo section in Jianhe County, Guizhou Province, located at 26°45.014′N latitude and 108°24.982′E longitude (Fig. 1).

## MATERIALS & METHODS

A total of 88 specimens (see File S1) are deposited at the Guizhou Research Centre for Palaeontology, Guizhou University. The specimen photos presented in this article were captured by Canon 5D Mark IV camera and VHX100K microscope. The resulting images were subsequently processed using the ImageJ (*Schneider, Rasband & Eliceiri, 2012*). The orientation of specimens during photography and measurements is primarily parallel with the cranidium, librigenae, or pygidium.

Not all disarticulated trilobite specimens are result from moulting behaviour, and non-exuviation factors such as transportation, biological disturbance, and decomposition during burial may also lead to exoskeleton disarticulation. In the present work, the exuviae identification follows the criteria established by *Henningsmoen (1975)*, *Whittington (1990)* and *Daley & Drage (2016)*, which primarily encompass: (1) The preservation of exuviae should consider the impact of sedimentary environments; (2) evidence shows suture opening; and (3) moult configurations exhibit a systematic, repetitive, and characteristic arrangement of structural units. According to the identification criteria, the exuviae of *Oryctocephalus indicus* were selected and classified into Henningsmoen's configuration, Harrington's configuration, and Somersault's configuration. This is the first report of the Somersault's configuration in *O. indicus*.

Some of the nomenclatures used to describe the studied material are listed below. The Axial shield (As) (Fig. 2B), is a single unit that includes the cranidium, thorax and pygidium (*Henningsmoen, 1975*; *Drage et al., 2018*). Thoracopygon (TP) (Fig. 2B), the thorax and pygidium are joined as a single unit (*Henningsmoen, 1975*; *Speyer, 1985*). Cephalothoracic

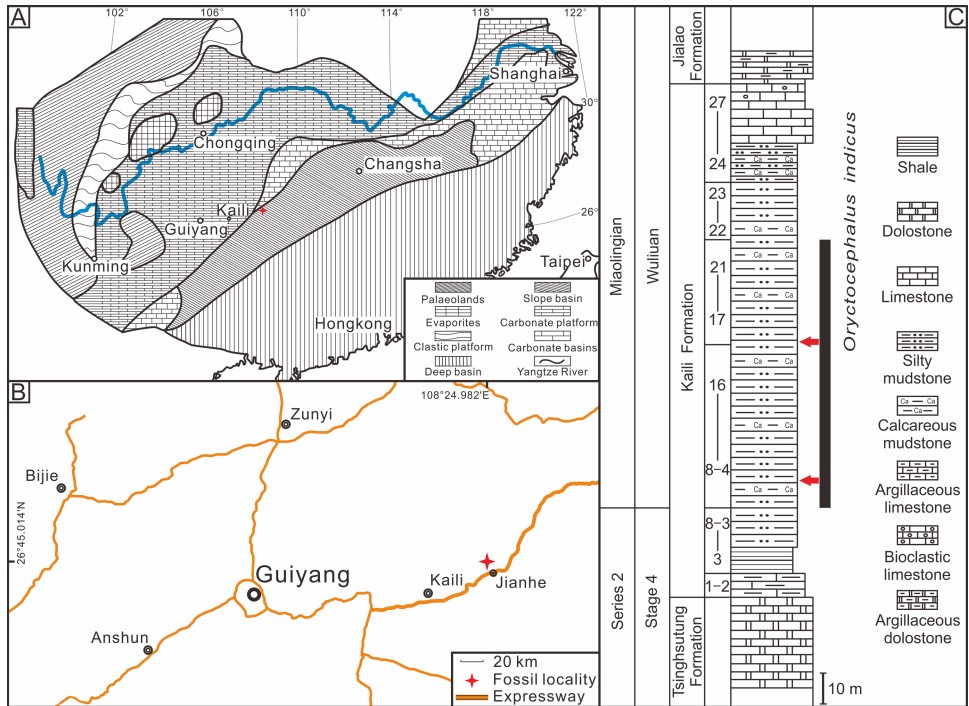

**Figure 1** **Location and stratigraphy of the Miaobanpo section.** (A) The lithofacies reconstruction of South China during the earliest Cambrian Miaolingian Series (modified from *Feng et al., 2004*). (B) A simplified sketch map of the fossil locality in Jianhe County, Guizhou Province, South China. (C) The Stratigraphicaln column of the Miaobanpo section (Kaili Formation) displays a horizon that yields *Oryctocephalus indicus* (modified from *Zhao et al., 2012*), the red arrows indicate the main collection horizon of the fossils in this study.

joint (CTJT) (Fig. 2B), articulation connecting the cephalon to the thoracopygon (*Drage, 2019*). Lower Cephalic Unit (LCU) (Fig. 2C), all parts of the cephalon except the cranidium, joined by integument or partially fused facial sutures (*Henningsmoen, 1975*; *Drage et al., 2018*). For other definitions of the nomenclatures (*e.g.*, moult configuration, moulting style and exuvial gape), refer to *Drage (2019)*, and references therein.

## RESULTS

### The moult configurations of *Oryctocephalus indicus*

Henningsmoen's configuration shows a slight displacement of the LCU and a cranidium displacement with regards to the thoracopygon (*Drage et al., 2018*). We found 80 specimens of Henningsmoen's configuration in *Oryctocephalus indicus*, which accounted for most of the material studied. According to the preservation characteristics of the exuviae, the moult specimens preserved in Henningsmoen's configuration are classified in this article into three types: Type I (consisting of seven specimens) showed an intact thoracopygon, the LCU slightly displaced and the cranidium was on the one side, some specimens (three specimens) displayed an inverted cranidium; Type II (consisting of 58 specimens) demonstrated that the thoracopygon remained connected, with a slight displacement of the LCU to connect

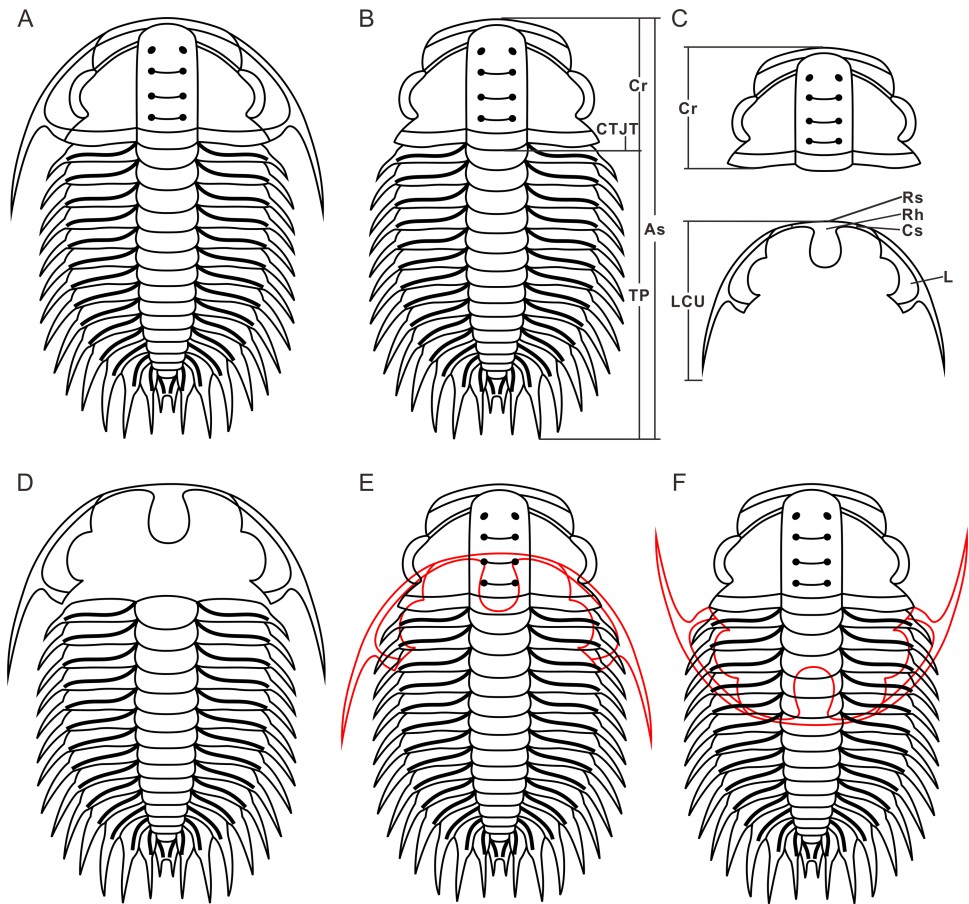

**Figure 2** **Reconstructions of *Oryctocephalus indicus*.** (A) Dorsal view of the exoskeleton. (B) Axial shield ($n = 2$). (C) Cranidium and lower cephalic unit. (D) Henningsmoen's configuration ($n = 80$). (E) Harrington's configuration ($n = 2$); (F) Somersault's configuration ($n = 4$). Structural units under the exoskeleton appear red. Abbreviations: As, Axial shield; Cr, Cranidium; TP, thoracopygon; LCU, lower cephalic unit; CTJT, cephalothoracic joint; L, librigenae; Cs, Connective suture; Rh, rostral-hypostomal plate; Rs, rostral suture.

with the thorax, and an absence of the cranidium; Type III (consisting of 15 specimens) is similar to Type II in the absence of the cranidium, but its LCU displacement is greater than that of Type II.

In Type I (Figs. 3A–3C), the most significant difference in Fig. 3C from the first two specimens (Figs. 3A and 3B) is the preserved position of the cranidium. In Fig. 3C, the cranidium is inverted so that its anterior end is in contact with the anterior end of the LCU. This observation would suggest that in some *Oryctocephalus indicus* individuals, the initial position of disarticulation may occur at the junction line between the cranidium and thorax, which is subsequently accompanied by splitting of the facial and rostral sutures to form the exuvial gape. Otherwise, if the disarticulation of the facial and rostral sutures completely separated the cranidium from the LCU, it is difficult to explain how the cranidium came to invert the front of the LCU and make contact with it. Nevertheless, this

is an exceptional case and comparable information is rarely found in other exuviae of *O. indicus*. Eventually, when the trilobite shed its old exoskeleton, the cranidium was inverted to the anterior end of the LCU. Due to the specimen fragmentation during field collection, only the cranidium (internal surface) and thoracopygon (external surface) were retained (Fig. 3B). The preservation state of this specimen was that the cranidium had flipped over and covered the right thorax. Supposing that the exuviae are not quickly buried after moulting, the cranidium inverted at the front of the LCU may also be transported (*e.g.*, Fig. 3A). Further disturbance could result in the absence of the cranidium, leading to Type II formation from Henningsmoen's configuration.

Most of the exuviae of *Oryctocephalus indicus* preserved in Henningsmoen's configuration were identified as Type II (Figs. 3D–3F), with a total of 58 specimens discovered during this study. In the complete specimens of *O. indicus*, the LCU is not connected to the thorax but instead attaches to the cranidium through the facial and rostral sutures. In Type II, the LCU was slightly posteriorly displaced and made contact with the first thoracic segment due to the absence of the cranidium (Figs. 3D–3F). The exuviae consists of a mineralised exoskeleton connected by the articulatory membranes and the unmineralised cuticle (*Whittington, 1990*). The presence of these articulatory membranes and the unmineralised cuticle well indicates that the LCU was preserved *in-situ* (or near-situ) and lay anterior to the thoracopygon after separating from the cephalon, unlike a case with the cranidium absent.

In the moult specimens of *Oryctocephalus indicus* collected for this study, Henningsmoen's configuration of Type III shows a more significant displacement of the LCU (Figs. 3G–3I). Judging from the preservation state of each structural unit of *O. indicus*, without considering the effect of moulting behaviour on the exuviae, it would appear that Type III in Henningsmoen's configuration (lacking cranidium and with LCU displacement) was affected more by non-moulting factors than Types I and II mentioned above. Specimen GTBM9-5-4110 (Fig. 3G) exhibits a lateral displacement of the LCU and absence of the cranidium, while the thoracopygon remains intact.

Based on the materials discovered thus far, the exuviae of *Oryctocephalus indicus* are mainly preserved in Henningsmoen's configuration, with a lesser number of exuviae being preserved in alternative configurations. One specimen shows *O. indicus* with the Axial shield (Fig. 4A), which is characterised by maintaining a connexion between the cranidium and the thoracopygon, yet lacking an LCU. Strictly speaking, the Axial shield is not a moult configuration *per se*, as it may be generated by other moult configurations (*Drage et al., 2018*). In many exuviae of *O. indicus* described here, the cranidium remains attached to the thoracopygon in *O. indicus* preserved in the Axial shield and Harrington's configuration, in contrast to Henningsmoen's configuration, in which the cranidium is separated from the thoracopygon. These specimens (Figs. 4B and 4C) show the preservation of *O. indicus* in Harrington's configuration, wherein the cranidium remains attached to the thoracopygon throughout the exoskeleton arrangement, and the LCU is posteriorly displaced under the exoskeleton following disconnexion between facial and rostral sutures.

So far, Somersault's configuration is identified for the first time from the *Oryctocephalus indicus* here. In this configuration, the librigenae are horizontally inverted so that the genal

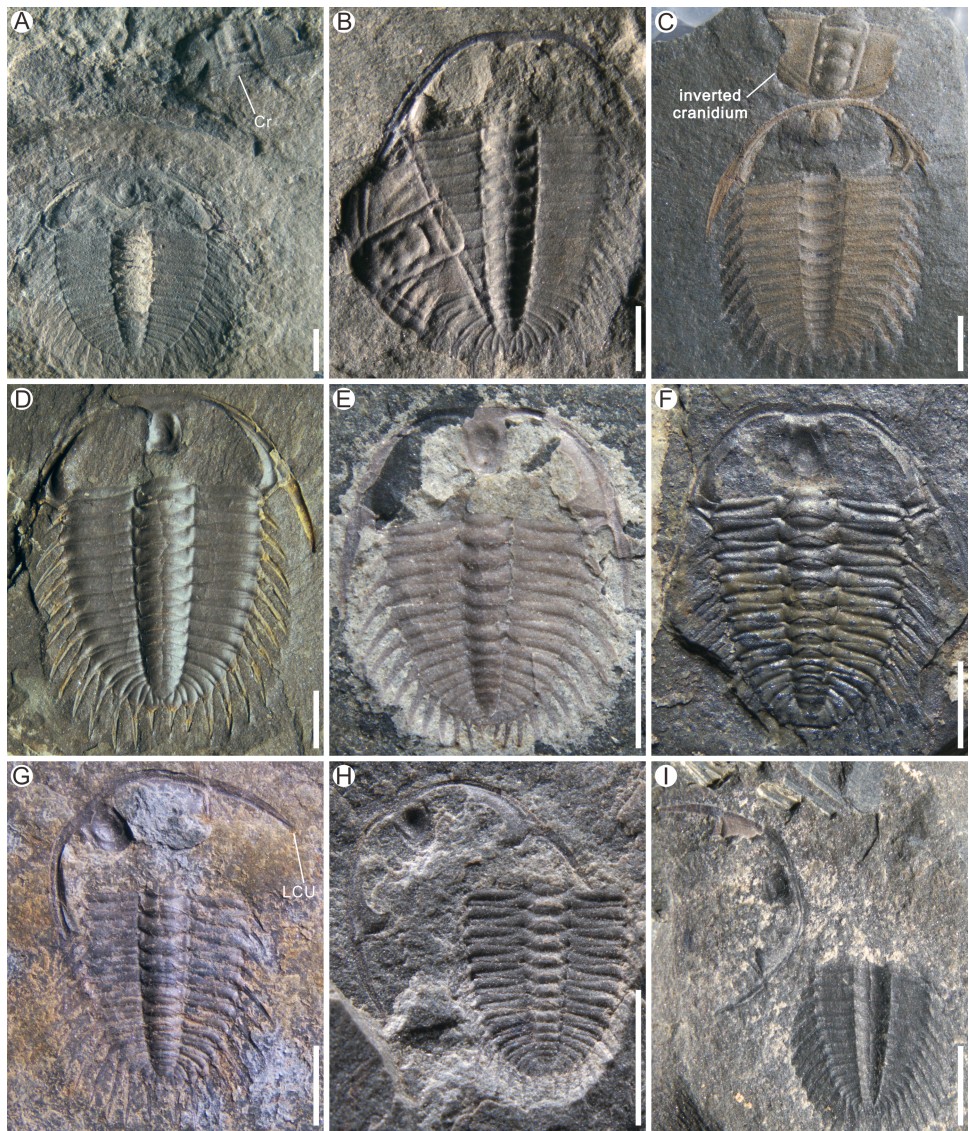

**Figure 3** **Examples of Henningsmoen's configuration in *Oryctocephalus indicus* from the Cambrian Kaili Formation of Guizhou Province, South China.** (A–C) Types I shows the preservation of cranidium (Cr); (B) Except for the cranidium is internal surface, the rest of the structures are external surface. (C) Shows the cranidium inverted before the LCU. (D–F) Types II shows the lower cephalic unit (LCU) nearly *in situ* and the displaced cranidium is missing; (G–I) Types III shows the laterally rotated of the LCU and the displaced cranidium is missing. Specimen Nos: A, GTBM17-1866; B, GTBM9-3-3706; C, GTBM16-437; D, GTBM9-3-3745; E, GTBM9-2-4282; F, GTBM9-5-4029; G, GTBM9-5-4110; H, GTBM8-5-832; I, GTBM9-3-2110. All scale bars are two mm.

spine points forward; if the rostral-hypostomal plate does not become disarticulated, it may be connected to the librigenae as an intact LCU (*Drage et al., 2018*). In specimen GTBM9-5-1024 (Fig. 4F), the LCU underwent a clockwise rotation of approximately 90° upon inversion and was subsequently positioned beneath the left side of the thorax, while the cranidium became detached from the thoracopygon and rotated 45° clockwise to face

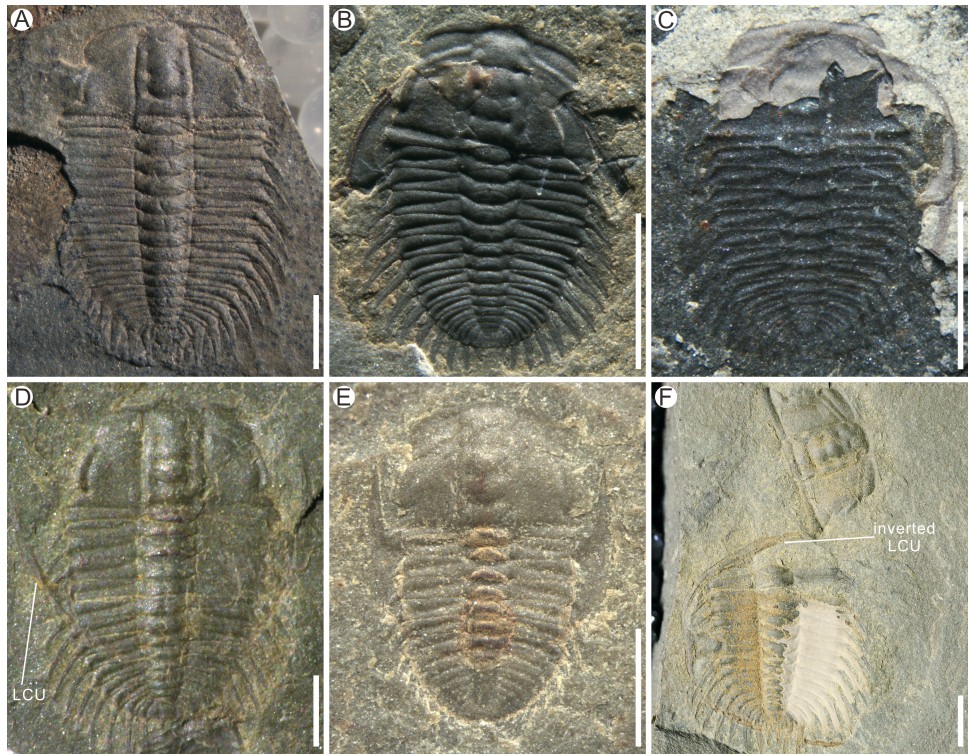

**Figure 4** **Examples of moult configurations in *Oryctocephalus indicus* from the Cambrian Kaili Formation of Guizhou Province, South China.** (A) Axial shield; (B–C) Harrington's configuration shows the backward displacement of the lower cephalic unit (LCU). (D–F) Somersault's configuration shows the inversion of the LCU and the absence or displacement of the cranidium. (E) shows the rostral-hypostomal plate (Rh) pressed beneath the thorax. Specimen Nos: A, GTBM9-2-1508; B, GTBM9-2-4316; C, GTBM9-1-752; D, GTBM9-2-2046; E, GTBM9-5-906; F, GTBM9-5-1024. All scale bars are two mm, except for E which is one mm.

anteriorly. Perhaps due to the specific nature of morphology in *O. indicus*, the cranidium was typically detached from the thoracopygon in Somersault's configuration (Fig. 4F), and only two specimens showing the cranidium remaining attached to the thoracopygon (Figs. 4D and 4E).

## The moulting style of *Oryctocephalus indicus*

Based on the exuviae of *Oryctocephalus indicus* preserved in Henningsmoen's configuration, Harrington's configuration, and Somersault's configuration, it is evident that the exuvial gape is created through shedding at least the LCU, and typically the cranidium. However, in the three moult configurations of *O. indicus*, there were slight differences in the disarticulation of the moult sutures. Perhaps influenced by biomechanics, Henningsmoen's configuration emphasises the involvement of the junction line between the cephalon and thorax in the moulting behaviour, where the LCU is *in-situ* mainly of the whole exoskeleton and is mostly unchanged by moulting activity. In others, the exuvial gape is formed primarily by disarticulation of the facial and rostral sutures, followed by occasional

displacement and/or inversion of the LCU due to the trilobite positioning during moulting and movement/biostratinomy.

## DISCUSSION

### Recognition of moult configurations in *Oryctocephalus indicus*

Describing patterns of ecdysis in the fossil record necessitates the ability to distinguish preserved exoskeleton moults from carcasses (*Daley & Drage, 2016*). With the development of trilobite research, numerous moult configurations have been proposed (*e.g.*, *Richter, 1937*; *Henningsmoen, 1957*; *Henningsmoen, 1975*; *Drage et al., 2018*), but few studies have addressed the formation of exuviae. The different moult configurations reflect a combination of moulting behaviour, movement of the trilobite during exuviation, and biostratinomic/preservational effects. In the following section, based on the previous description of moult configurations, we will provide further explanations for the moulting behaviour of *Oryctocephalus indicus*.

In Henningsmoen's configuration, trilobites can expand their exuvial gape by shedding the cranidium from the thoracopygon (*Drage et al., 2018*; *Corrales-García et al., 2020*; *Wang et al., 2020*). It is yet to be determined whether this strategy was consciously chosen by *Oryctocephalus indicus*, perhaps the difficulty of moulting forces the trilobite to generate more movement, resulting disarticulation and displacement of the cranidium during moulting. The significance of this configuration for *O. indicus* in our research is to underscores the pivotal role played by the opening at the cephalothoracic joint, as well as the facial and rostral sutures, in the formation and enlarging of the exuvial gape. We propose that specimens, only exhibiting cranidium displacement or absence but with relatively intact structural units and minimal displacement, should be classified as exuviae rather than products of physical disturbance. Little is known about the burial of most specimens considered to be moult configurations (*Speyer, 1985*; *Speyer, 1987*; *Whittington, 1990*). *Wang et al. (2020)* illustrated a series of changes in the exuviae of *Arthricocephalites xinzhaiheensis* after disturbance. Similarly, the exuviae of *O. indicus* preserved in Henningsmoen's configuration reported in this article shows increasing amounts of disruption or transport (either by abiotic disturbance or by movement of the trilobite during moulting), and we have divided them into three types. Type I of Henningsmoen's configuration represents less disturbance, such as currents, during exuviae preservation at the end of moulting behaviour. Specimen GTBM16-437 (Fig. 3C) displays an unusual arrangement where the cranidium is inverted and comes in contact with the anterior end of the LCU, as illustrated by exoskeleton portions in the moult configuration. This suggests that this arrangement was not accidental during transport and burial. As previously mentioned, in the exuviae of *O. indicus* preserved in Type II of Henningsmoen's configuration (Figs. 3D–3F), the presence of intact LCU and thoracopygon refutes the case that the cranidium was absent due to intense transport. Without considering the possibility that the trilobite carried the cranidium away during moulting, the absence of the cranidium in Type II compared to Type I seems to indicate that the exuviae may have been disturbed before burial. In Type III of Henningsmoen's configuration (Figs. 3G–3I), it remains
uncertain whether the displacement of LCU was solely attributed to moulting activity, and to what extent these particular specimens were affected by burial. However, solely based on the state of preservation and location of the structural units, it is undisputed that Type III is disrupted the most among the three types in Henningsmoen's configuration.

The formation of the exuvial gape is related to the disjunction of various sutures, which are influenced by the distinct structural properties of trilobites. Therefore, when discussing the moulting behaviour of different trilobites, overall structural features must be taken into consideration. To a certain extent, Harrington's configuration bears resemblance to specimens in which the disarticulation of the librigenae was affected by burial. Noteworthily, in the Harrington's configuration exhibited by the exuviae of *Oryctocephalus indicus*, only the LCU is displaced while the librigenae remains connected with the rostral-hypostomal plate. The consistency of Harrington's configuration suggests that the formation of this configuration was due to the moulting behaviour of the trilobites (*Henningsmoen, 1975*). When the structural units, such as the cranidium, thorax, and pygidium, are fully preserved, it becomes challenging to attribute displacement of the LCU solely to non-moulting factors. The exuviae of trilobite preserved in Harrington's configuration has also been reported in several previous papers (*e.g.*, *Harrington & Leanza, 1957*; *Henningsmoen, 1957*; *Henningsmoen, 1975*; *Chatterton & Ludvigsen, 1998*; *Chen, Han & Zhao, 2008*; *Ebbestad et al., 2013*; *Drage et al., 2018*). The exuviae of *Estaingia bilobata* from the Emu Bay Shale, South Australia, shows remarkable similarity in moult configuration with *O. indicus*, as the disarticulation and posterior displacement of the LCU relative to the cranidium were also exhibited in *E. bilobata* preserved in Harrington's configuration (*Drage et al., 2018*). It is obvious that the entire movement of the LCU after disarticulation from the cephalon plays a specific and positive role in the enlargement of the exuvial gape.

Previous research has indicated that preservation of trilobite exuviae exhibiting Somersault's configuration is uncommon, as it necessitates a strong dorsal flexure during the moulting process and prompt burial after exuviation (*Whittington, 1990*; *Drage et al., 2018*; *Wang et al., 2020*). In Somersault's configuration observed in the exuviae of *Eosoptychopara guizhouensis*, the cranidium usually remains connected to the thoracopygon while the LCU is inverted and overlain under the thorax (*Chen et al., 2022*; Fig. 3). The prevalence of Henningsmoen's configuration in exuviae of *Oryctocephalus indicus*, characterised by the absence of the cranidium, suggests that the junction between the cephalon and thorax is prone to breakage during moulting activity, resulting in some exuviae preserved in Somersault's configuration exhibiting separation of the cranidium and thoracopygon. The weakest link in exuviae is between the cranidium and the first thoracic segment, and the cranidium might have been dislocated, inverted, or carried away from the remainder of the exuviae in the course of burial (*Whittington, 1990*). Therefore, the structural characteristics of *O. indicus* specimens (Fig. 4F) that show separation between the cranidium and thoracopygon are slightly different from Somersault's configuration documented in previous studies.

Based on a large sample of exuviae from meraspid to holaspid, *Wang et al. (2021)* suggested that there is a gradual transition in the ontogenetic moulting behaviour of *Arthricocephalites xinzhaiheensis*. In the early stages of ontogeny, the exuviae of

*Ar. xinzhaiheensis* only exhibits Somersault's configuration, whereas in the later stages Henningsmoen's configuration dominates (*Wang et al., 2021*). To figure out the question whether the moulting configuration of exuviae would change due to ontogenesis, 88 exuviae of *O. indicus* are measured and obtained the results shown in Fig. 5. Collecting bias for 'display-quality' specimens may also affect interpretations of moulting behaviour variability for trilobites (*Drage et al., 2018*). The exuviae of smaller individuals seem to be less likely to be found and collected during fieldwork, so it should be pointed out that the following results are merely obtained based on the current database. According to the proportion of moulting configuration in Fig. 5A, in the case of *O. indicus* preserved in Henningsmoen's configuration, except for one exuviae in M7 (M designates meraspid instars), the rest are mainly from the late meraspid and holaspid stages. The discovery of numerous exuviae preserved in Henningsmoen's configuration in the late meraspid and holaspid stages seems to indicate that the possibility of enlarging the exuvial gape by disarticulating the sutures of the cephalothoracic joints gradually increases as the trilobite approaches the holaspid stage. Concerning the correlation between exuviae size and the diversity of moulting configurations in *O. indicus*, the Axial shield, Harrington's configuration, and Somersault's configuration are primarily present in the 0–8 mm range, whereas in the 8–14 mm range all exuviae are preserved in Henningsmoen's configuration (Fig. 5B). This result also further illustrates the gradual dominance of Henningsmoen's configuration with increasing trilobite size in the exuviae of *O. indicus.*

## The putative moulting process of *Oryctocephalus indicus*

The repeated occurrence of particular exoskeletal configurations and their agreement with inferred ecdysial openings can reveal how exuviation occurred in a given species (*Henningsmoen, 1975*). After the preceding discussion of the moult configurations and style of *Oryctocephalus indicus* preserved in Henningsmoen's configuration, Somersault's configuration, and Harrington's configuration, we present a preliminary analysis of the formation process of these three configurations based on information from the exuviae. The information provided by the Axial shield regarding the moulting process is limited, and it is possible that this structure formed subsequent to other moult configurations being transported. Therefore, there is no discussion of the formation process of the Axial shield in this context.

In the detailed discussion of moult configurations above, we have observed that Henningsmoen's configuration has highlighted the importance of the junction line between cephalon and thorax in moulting behaviour. However, among the exuviae of *Oryctocephalus indicus* in other moult configurations, only some of the exuviae preserved in Somersault's configuration show the cranidium disarticulated from the thoracopygon. The distinct processes of formation exhibited by moulting specimens of *O. indicus* in Henningsmoen's configuration are depicted (Fig. 6B). Here, we refer to the moulting process proposed by *Whittington (1990)*, in which trilobites bend themselves to open the suture and create an exuvial gape, which subsequently results in different moult configurations due to discrepancies in moulting behaviour. As mentioned above, we classified the exuviae of *O. indicus* preserved in Henningsmoen's configuration into three types based on the

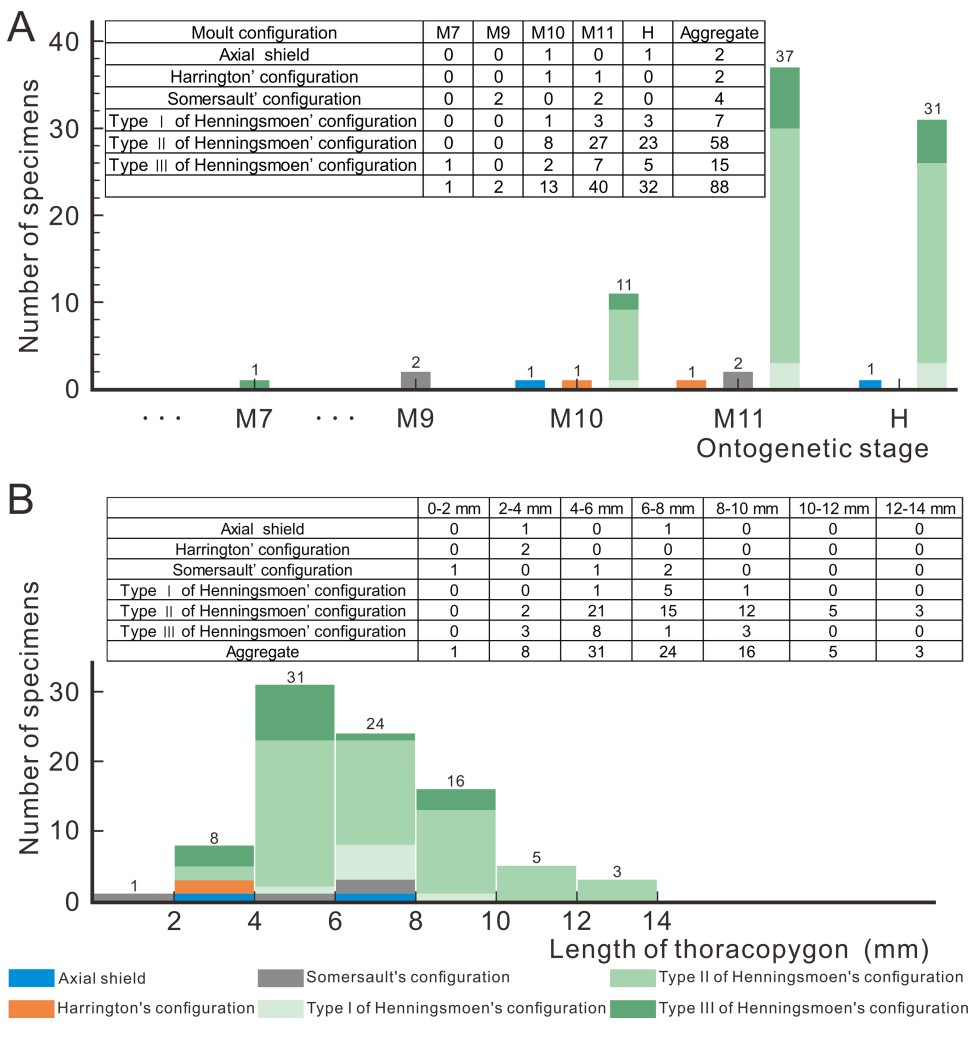

**Figure 5  Moult configurations of *Oryctocephalus indicus* in relation to ontogenetic stage and individual size.** (A) The number of specimens at different ontogenetic stages; (B) the size distribution of thoracopygon length. Abbreviations: M designates meraspid instars and H designates holaspid instars (see File S1 for details).

preservation status of the cranidium and LCU. Figure 6B illustrates the formation of a particular case in Type I of Henningsmoen's configuration. Perhaps the initial suture that opened during moulting was located at the cephalothoracic joint, and then, the exuvial gape was further expanded by the disconnexion of both facial and rostral suture due to moulting activity. Finally, the trilobite crawls out of its old exoskeleton while flipping the cranidium so that its front area intersects with the front end of the rostral-hypostomal plate (Fig. 6B). Such examples with the cranidium inverted at the anterior end of the LCU are rare, and other exuviae of *O. indicus* preserved in the Type I of Henningsmoen's configuration only show the cranidium is displaced. Therefore, in the exuviae of *O. indicus* preserved in Henningsmoen's configuration, the exuvial gape is generally formed by disarticulation of the facial and rostral sutures, and then the cranidium is separated from the thoracopygon

during the movement of the exuviating individual. Then, the trilobite crawled out of its old exoskeleton through the exuvial gape, displacing the cranidium and leaving behind the LCU and thoracopygon. The difference between Figs. 6B and 6C lies in the absence of the cranidium in Figs. 6C, whereas 6D not only lacks the cranidium but further shows the displacement of the LCU. The moult configuration shown in Figs. 6C and 6D in a particular environment could also have occurred after displacement of the cranidium or LCU, which shows this possibility with a dashed line (Fig. 6B).

The exuviae of *Oryctocephalus indicus* in Somersault's configuration and Harrington's configuration have certain similarities during the moulting process, both of which completed the moult through the exuvial gape created by the separation of LCU from cephalon, but their divergent moulting behaviour resulted in distinct moult configurations. In the exuviae of *O. indicus* preserved in Somersault's configuration (Fig. 6E), disconnexion between facial and rostral sutures results in a separation between the LCU and cranidium, creating an exuvial gape in front of the cephalon for ecdysis. When the trilobite crawled out of its old exoskeleton, it caused the LCU to move in the direction indicated by the arrow and ultimately assumed Somersault's configuration, in which its thorax and pygidium connected while the LCU was inverted and pressed under the thorax (Fig. 6E). The cranidium was easily separated from the thorax in *O. indicus*, resulting in a significant absence or displacement of most of the cranidium in the Somersault's configuration. However, we hypothesise that Harrington's configuration observed in *O indicus* (Fig. 6F) may be attributed to insufficient dorsal flexure during the moulting process, and that the moulting behaviour of the trilobite failed to cause the LCU to overturn after breaking the facial and rostral suture. Instead, the LCU was pushed backwards by the forward movement of the trilobite, eventually forming Harrington's configuration in which the cranidium, thorax and pygidium were connected, while the LCU was displaced below the thorax.

## CONCLUSIONS

Based on the established criteria for identifying exuviae in previous studies and the detailed description of *Oryctocephalus indicus* provided in this article, we propose that the disarticulated specimens presented herein are trilobite exuviae resulted from moulting behaviour rather than simple mechanical transport. The exuviae of *O. indicus* are predominantly preserved in Henningsmoen's configuration, with some specimens exhibiting Harrington's configuration or Somersault's configuration. Additionally, we have categorised Henningsmoen's configuration into two types (Type I and Type II) based on the presence or absence of the cranidium in the exuviae, and further distinguished Type III from Type II by examining whether or not the LCU is displaced.

In the exuviae of *Oryctocephalus indicus*, the LCU was detached from the cephalon and positioned anterior to the thoracopygon, and the majority of exuviae show that the cranidium was absent. In this article, we provide a detailed discussion on Henningsmoen's configuration based on the moulting information supplied by the exuviae of *O. indicus*, and suggest that disturbance prior to complete burial may be one of the main reasons for the change in Henningsmoen's configuration. Simultaneously, while endeavouring

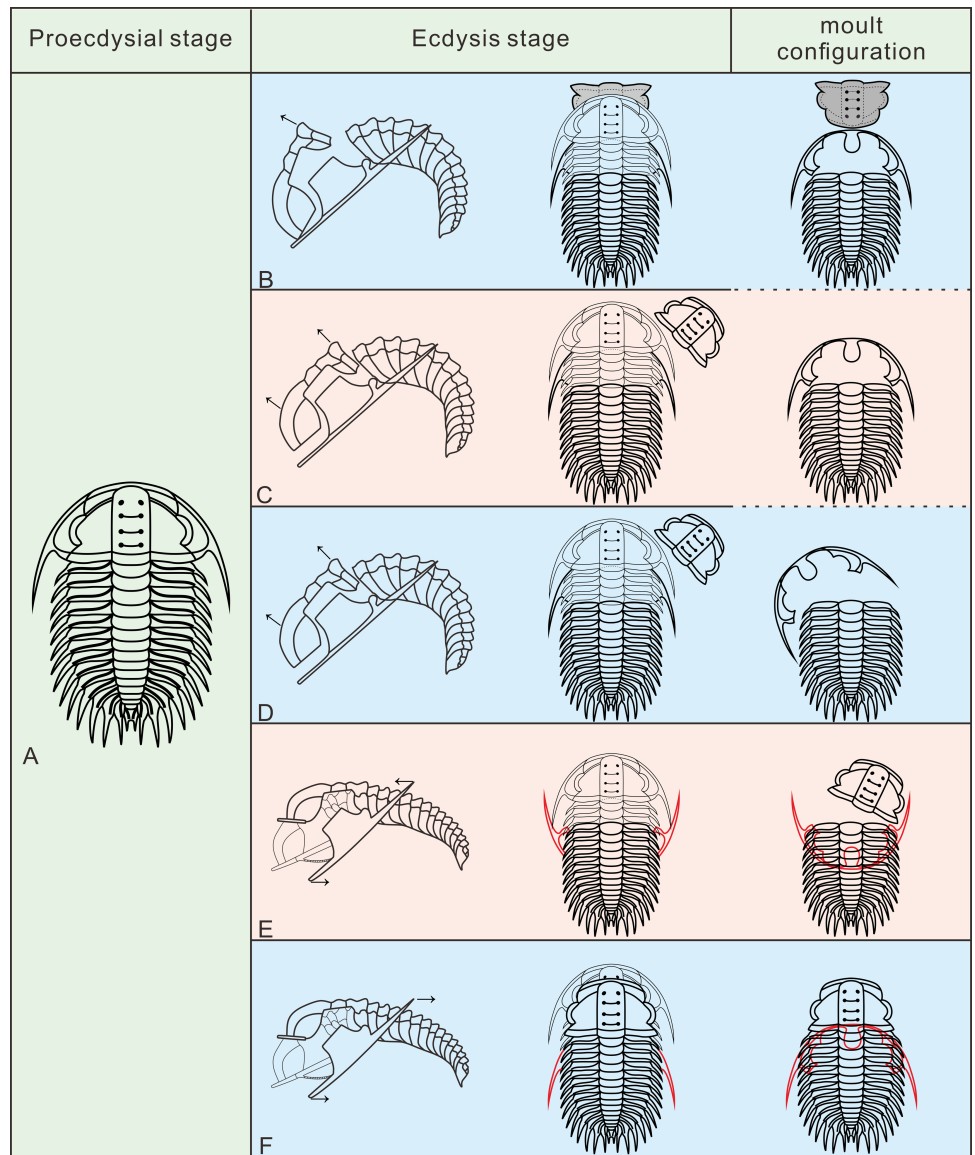

**Figure 6  Reconstruction of moult process for *Oryctocephalus indicus* based on specimens.** (A) A complete individual at the pre-ecdysial stage; (B, C), and (D) is the formation processes of three different types in Henningsmoen's configuration, the inverted cranidium in $B_1$ is shown in grey; (E) the formation process of Somersault's configuration; (F) the formation process of Harrington's configuration. After inverted or displacement, structural elements under the thoracopygon are shown in red, and trilobites in the moulting process are plotted in thin lines. The moulting process was modified from *Whittington (1990)*.

to reconstruct the moulting process of *O. indicus*, we suggested that differences in the moulting process (*e.g.*, degree of dorsal flexure, whether the exuvial gape is enlarged by detaching the cranidium, *etc.*) may lead to changes in moulting configurations. With the information gained from studying the moulting behaviour of *O. indicus*, we believe that even for trilobites preserved in recognised moulting configurations, it is imperative to

consider whether non-moulting factors have affected the preservation state of structural units, in the exuviae prior to complete preservation.

## ACKNOWLEDGEMENTS

We are grateful to Yuanlong Zhao, who has made great contributions to the palaeontology of Guizhou Province, and this article has benefited from his guidance. Harriet B. Drage (Institute of Earth Sciences, University of Lausanne) and an anonymous reviewer are thanked for their constructive review comments, which helped to improve the manuscript. We thank Yifan Wang for the discussion. Feng Liu and Zefu Liu helped with the fieldwork.

### Funding

This work was supported by the National Natural Science Foundation of China (grant numbers 42330209, 42262003, 42162005), the Guizhou Bureau of Science and Technology (grant numbers Gui. Sci. Sup. [2020]4Y241; Gui. Sci. Tal. [2017] 5788). The funders had no role in study design, data collection and analysis, decision to publish, or preparation of the manuscript.

### Grant Disclosures

The following grant information was disclosed by the authors:
National Natural Science Foundation of China: 42330209, 42262003, 42162005.
Guizhou Bureau of Science and Technology: Gui. Sci. Sup. [2020]4Y241, Gui. Sci. Tal. [2017] 5788.

### Competing Interests

The authors declare there are no competing interests.

### Author Contributions

- Shengguang Chen conceived and designed the experiments, performed the experiments, analyzed the data, prepared figures and/or tables, authored or reviewed drafts of the article, and approved the final draft.
- Xinglian Yang conceived and designed the experiments, performed the experiments, analyzed the data, prepared figures and/or tables, authored or reviewed drafts of the article, and approved the final draft.
- Xiong Liu conceived and designed the experiments, performed the experiments, analyzed the data, authored or reviewed drafts of the article, and approved the final draft.
- Zhengpeng Chen conceived and designed the experiments, performed the experiments, analyzed the data, prepared figures and/or tables, and approved the final draft.
- Zhixin Sun conceived and designed the experiments, performed the experiments, analyzed the data, authored or reviewed drafts of the article, and approved the final draft.

- Fangchen Zhao conceived and designed the experiments, performed the experiments, analyzed the data, prepared figures and/or tables, authored or reviewed drafts of the article, and approved the final draft.

## Data Availability

The raw data are photographs in Fig. 1–6.

All the specimens are deposited in the Guizhou Research Centre for Palaeontology, Guizhou University, Guizhou, China: GTBM17-1866, GTBM9-3-3706, GTBM16-437, GTBM9-3-3745, GTBM9-2-4282, GTBM9-5-4029, GTBM9-5-4110, GTBM8-5-832, GTBM9-3-2110, GTBM9-2-1580, GTBM9-2-4316, GTBM9-1-752, GTBM9-2-2046, GTBM9-5-906, GTBM9-5-1024.

## Supplemental Information

Supplemental information for this article can be found online at http://dx.doi.org/10.7717/peerj.16440#supplemental-information.

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
