# Peer review of "Moulting behaviour in the trilobite Oryctocephalus indicus (Reed, 1910) from the Cambrian Miaolingian Series (Wuliuan Stage) of Jianhe, South China"

_PeerJ, doi:10.7717/peerj.16440_

## Round 0.1 · original submission · Major Revisions

Both reviewers have raised serious concerns and, therefore, your paper cannot be accepted for publication in PeerJ. However, since the reviewers do find some merit in the paper, I would be willing to reconsider if you wish to undertake major revisions, addressing the referees' concerns.

Please note that resubmitting your manuscript does not guarantee eventual acceptance, and your resubmission will be subjected to re-review before a decision is rendered.

·

Basic reporting

Note: the majority of my comments and queries are in the attached PDF.

• It doesn’t seem that the specimen numbers provided cover all of the dataset used for this study (only 15 numbers given, compared to 88 exuviae analysed?) – please confirm they do, or provide specimen numbers for all specimens used for the study.

• The figures are good, and I think the specimens chosen generally show the moult configuration types well.
Figure 5: are the exuviation drawings in B1 and B2 meant to have the moulting trilobite present within the old exoskeleton (like C and D do)?

• Some of the grammar could be improved throughout, as the lack of precision makes certain parts difficult to interpret (e.g., description of the Henningsmoen’s configuration types). However, I’ve only noted these issues where they change the meaning of the scientific content.

• The figure callouts are not always in alphabetical order in the text (if this is something PeerJ requires).

Experimental design

Note: the majority of my comments and queries are in the attached PDF.

• Are all specimens adults? Given moulting behaviours can change with ontogeny (e.g., Drage et al., 2018, Paleobiology).

• Is there any observable difference in size between the specimens in each of the Henningsmoen’s configuration type categories? Or between the different moult configurations? I know the other configuration types are rare, but it would be interesting if they happened to be of a notably different body size to other moult configurations.

Validity of the findings

Note: the majority of my comments and queries are in the attached PDF.

• A lot of my comments in the PDF relate to the interpretation of the moulting processes that produce moults in Henningsmoen’s configuration. I don’t see evidence for the interpretation that the cephalothoracic joint was disarticulated first in this species, given the species clearly has functional facial sutures, and these are ubiquitously used for moulting across Trilobita (and likely existed purely to aid this behaviour). I think that it is more likely that this species had a cephalothoracic joint that was prone to disarticulation during moulting – this could be because the joint in this species was particularly weak, or the individuals often put lots of pressure on it (e.g., by severe dorsal flexure or movements during moulting), or they had a particular morphological feature that basically required this joint to break to enlarge the exuvial gape (e.g., maybe a particularly broad thorax). I don’t know the answer to this (and I don’t think it is necessarily possible to make concrete decisions about these aspects), but I think this needs addressing throughout the manuscript.

• Would you say that the three types of Henningsmoen’s configuration therefore represent a taphonomic spectrum, with the actual type preserved in the fossil record being almost entirely influenced by biostratinomy? Perhaps even a pathway, with increasing amounts of disruption/transportation (either abiotic or due to movement of the trilobite) prior to preservation as we go from type 1 to 3? This seems likely to me, and I think it’s really interesting that you could identify such a thing in your sample. I would suggest adding text to discuss this idea in the discussion section.

• I don’t think that calling it ‘the reliability of O. indicus moult configurations’ really makes sense in light of the content of the paper.

Additional comments

Thanks very much to the editor and authors for sharing this paper with me, and suggesting me as a peer reviewer.

The study is based on a large number of well-preserved trilobite moult specimens, which therefore allows in depth investigation of the intraspecific variability of moulting and the moult configurations preserved in a single species. The figuring and interpretation of the moult specimens is reasonable, and this work provides a further useful example of detailed moulting information in trilobites.

I have some questions and comments related to the broader considerations of the paper, which should be addressed before I would recommend publication. In particular, some of these are suggestions for further aspects to discuss on moulting and preservation of trilobite moults, because the dataset presented here raises some interesting considerations. Most of the comments I have made are within the attached review PDF, and I have added a few supplementary comments.

I hope this review is useful to the authors!

All the best,
Harriet B. Drage

Reviewer 2 ·

Basic reporting

Chen et al. described intraspecific diversity in the molt configuration of Cambrian trilobite Oryctocephalus indicus and identified three distinct configurations. Intraspecific diversity of molting has been identified in many trilobites, and the three molt configurations described in this paper are also present in other species. This paper is only a new case study, and no new molt configurations or strategies are proposed. After reading through the whole paper, almost all of this paper describes the molt mode of Oryctocephalus indicus and discusses its molt process. Although the description and discussion are sufficient and reasonable, I am still worried about the innovation and highlights of this paper, and perhaps it would be better suited to a specialized journal of paleontology that prefers systematic descriptions. A salvage measure for the authors to consider, you have collected a lot of specimens, why not do some statistical work? For example, whether the diversity of molt configuration is related to individual sizes, there is only one molt configuration in early stage, but it increases to three in the late stage. In addition, it can be seen from Figure 1 that these trilobites were collected from two different horizons. Is there any difference in the diversity of molt configuration between these two horizons? You do not provide this information in the paper. If so, could the diversity of molt configuration be related to the sedimentary environment? The two different horizons in Figure 1 may represent different sedimentary environments. I hope the author can consider these suggestions and add related work, and maybe get new understanding.

Experimental design

No comments.

Validity of the findings

No comments.

Additional comments

I made some accompanying comments in the PDF.

Annotated reviews are not available for download in order to protect the identity of reviewers who chose to remain anonymous.

---

## Round 0.2 · Minor Revisions

There are quite a few grammatical errors in the paper, including line 54, 59, 87, 91, 92, 143, 147, 275, etc. Also, it is better not to mix American English and British English in one paper. Please check the manuscript thoroughly.

**Language Note:** The Academic Editor has identified that the English language must be improved. PeerJ can provide language editing services - please contact us at copyediting@peerj.com for pricing (be sure to provide your manuscript number and title). Alternatively, you should make your own arrangements to improve the language quality and provide details in your response letter. – PeerJ Staff

Reviewer 2 ·

Basic reporting

I am glad to see that the authors have made substantial changes to the manuscript. The authors seem to have confused the identity of the reviewers with the PDF attachment, but this doesn't matter much because the authors responded to all the comments. I found a few more small problems, and after solving them I think the manuscript is ready for publication.

Line 71-73, the latest reference was published in 2011. I know that there are still papers on the moulting process of trilobite published recently. Please add some newer papers. Such as: Drage, H.B. Holmes, J.D., García-Bellido, D.C. & Daley, A.C. 2018: An exceptional record of Cambrian trilobite moulting behaviour preserved in the Emu Bay Shale, South Australia. Lethaia, Vol. 51, pp. 473–492.

Line 82, for "horizontal lamina", do you mean “horizontal bedding”? Please correct it.

Line 83, for "disturbed structures," you mean “bioturbated structures”?

Lines 110 and 142, for "IV" and "I", keep them in the same font (Times New Roman) as the rest of the text, and check the other numbers.

Experimental design

No comment.

Validity of the findings

No comment.

Additional comments

No comment.

---

## Round 0.3 · accepted · Accept

Thank you for having addressed all of the reviewers' comments. I have considered your revised manuscript and recommended publication in PeerJ. We are pleased to accept your paper in its current form.